



**The atmospheric fate of 1,2-Dibromo-4-(1,2-dibromoethyl)cyclohexane (TBECH):**
**Spatial patterns, seasonal variability, and deposition to Canadian coastal regions**
Jenny Oh,[1,2] Chubashini Shunthirasingham,[3] Ying Duan Lei,[1] Faqiang Zhan,[1] Yuening Li,[1] Abigaëlle Dalpé
Castilloux,[4] Amina Ben Chaaben,[4] Zhe Lu,[4] Kelsey Lee,[5] Frank A. P. C. Gobas,[5] Sabine Eckhardt,[6] Nick
Alexandrou,[3] Hayley Hung,[3] Frank Wania[1,2,*]
*[1] Department of Physical and Environmental Sciences, University of Toronto Scarborough, 1265 Military*
*Trail, Toronto, Ontario, Canada M1C 1A4*
*[2] Department of Chemistry, University of Toronto Scarborough, 1265 Military Trail, Toronto, Ontario,*
*Canada M1C 1A4*
*[3] Environment and Climate Change Canada, Downsview, 4905 Dufferin St, North York, Ontario, Canada*
*M3H 5T4*
*[4] Institut des Sciences de la Mer de Rimouski, Université du Quebec à Rimouski, 300 allée des Ursulines,*
*Rimouski, Québec, Canada G5L 3A1*
*[5] Simon Fraser University, 8888 University Dr, Burnaby, British Columbia, Canada V5A 1S6*
*[6] Norwegian Institute for Air Research, Instituttveien 18, 2007 Kjeller, Norway*
*Corresponding author: frank.wania@utoronto.ca



## Abstract

Brominated flame retardants (BFRs) that are gradually phased out are being replaced by emerging BFRs.
Here, we report the concentration of the α- and β-isomers of 1,2-Dibromo-4-(1,2-dibromoethyl) cyclo–
hexane (TBECH) in over 300 air, water, and precipitation samples collected between 2019 and 2022 using
active air and deposition sampling as well as networks of passive air and water samplers. The sampling
region includes Canada's most populated cities and areas along the St. Lawrence River and Estuary,
Quebec, as well as around the Salish Sea, British Columbia. TBECH was detected in over 60% of air samples
at levels comparable to those of 2,2',4,4'-tetrabromodiphenyl ether (BDE-47). Concentrations of TBECH
and BDE-47 were typically higher in urban areas, with stronger correlations with population density during
warmer deployments. Uniform α/β-TBECH ratios across space, time and environmental media indicate
highly similar atmospheric fate of the two isomers. Although TBECH air concentrations were strongly
related to temperature in urban Toronto and a remote site on the East coast, the lack of such dependence
at a remote site on the West coast can be explained by the small seasonal temperature range and
summertime air mass transport from the Pacific Ocean. Despite there being no evidence that TBECH has
been produced, or imported for use, in Canada, it is now one of the most abundant gaseous BFRs in the
Canadian atmosphere. The recorded spatial and temporal variability of TBECH suggest that its emissions
are not constrained to specific locations but are generally tied to the presence of humans. The most likely
explanation for its environmental occurrence in Canada is the release from imported consumer products
containing TBECH. Chiral analysis suggests that despite its urban origin, at least some fraction of TBECH
has experienced enantioselective processing, i.e., has volatilized from reservoirs where it has undergone
microbial transformations. Microbial processes in urban soils and in marine waters may have divergent
enantioselectivity.



## 1. Introduction


Starting from the 1970s, brominated flame retardants (BFRs) have been used extensively in commercial
products to reduce ignition and increase their resistance to fire. To date, at least 75 different kinds of BFRs
have been commercially produced, with polybrominated diphenyl ethers (PBDEs) being historically the
most used class (Alaee et al., 2003). However, their toxicity, potential for bioaccumulation, and
persistence in the environment soon became apparent (Christensen et al., 2005; de Wit et al., 2006; Palm
et al., 2002; Ruan et al., 2019), and PBDEs, along with other traditional BFRs, were eventually classified as
persistent organic pollutants (POPs) (POPRC, 2009). As usage and production of these BFRs were restricted
internationally in 2009 under the United Nations Environment Programme Stockholm Convention on
POPs (UNEP-SC-POPs) (UNECE, 2018), emerging brominated flame retardants (EBFRs) entered into the
market (Betts, 2008), which are touted to be less persistent than their traditional BFR counterparts.
However, studies on their environmental behaviours have been limited.
1,2-Dibromo-4-(1,2-dibromoethyl)cyclohexane (TBECH) is an EBFR that is widely used in plastics, fabric
adhesives, and building insulation materials (Alaee et al., 2003). TBECH exists as four diastereomers (α, β,
γ, and δ-TBECH). Furthermore, TBECH exhibits chirality and thus each diastereomer consists of a pair of
enantiomers. While commercial TBECH comprises mostly the α- and β-isomers, TBECH has been reported
to thermally isomerize to the γ and δ isomers when heated to 123 °C (Arsenault et al., 2008). Although
global production volumes or emissions of TBECH are difficult to establish, reports on its detection in the
environment have been increasing over the years, attributed to the rising demand for EBFRs (Bohlin et al.,
2014; Cequier et al., 2014; Drage et al., 2016; Genisoglu et al., 2019; Gentes et al., 2012; Newton et al.,
2015; Pasecnaja et al., 2021; Tao et al., 2017; Zacs et al., 2021). TBECH was first detected in sediment
samples collected in 1996 near a discharge pipe of the Frutarom plastics plant near Haifa, Israel (Santillo
et al., 1997). Since then, TBECH was found in various environmental media worldwide, such as biota and
water, with α- and β-TBECH being the dominant isomers. TBECH concentrations appear to be particularly
high in indoor settings, such as offices and residences (Melymuk et al., 2016; Newton et al., 2015; Wong
et al., 2018).
There is evidence that TBECH displays toxic effects at low concentrations both *in vitro* and *in vivo*. All
TBECH isomers have shown to be strong androgen agonists, where TBECH competitively binds to the
androgen receptor active site in varying degrees, first determined *in vitro* in human cell lines (Khalaf et
al., 2009; Larsson et al., 2006). Moreover, several studies suggest that all isomers are capable of



multimodal endocrine disruption which involves estrogenic (Asnake et al., 2014), anti-androgenic (Wong
et al., 2016), and thyroidal processes (Porter et al., 2014). This disrupting potential of TBECH is also seen
in the *in vivo* studies reporting changes in circulating hormones (Curran et al., 2017; Gemmill et al., 2011),
organ structure (Park et al., 2011), and reproduction (Marteinson et al., 2012a; Marteinson et al., 2012b)
after exposure to low concentrations of TBECH.
Presumably because no Canadian company reported the manufacture and importation of TBECH on a
commercial scale (ECCC, 2017), it has been deemed a non-priority for assessment under Canada's
Chemicals Management Plan (ECCC, 2019). As a result, TBECH remains unregulated at both national and
international levels, despite being judged to be persistent and bioaccumulative. There is evidence,
however, of its presence in the Canadian environment, as TBECH has been detected in different
environmental media, ranging from urban outdoor air in Toronto (Shoeib et al., 2014) to whale blubber
in the Arctic (Tomy et al., 2008).  Only three studies have published on the enantiomeric composition of
TBECH in the environment (Ruan et al., 2019; Ruan et al., 2018), with one study on Canadian urban soils
(Wong et al., 2012). Therefore, concerted efforts to understand the sources, atmospheric behaviour, and
enantiomeric profile of TBECH have been limited, not only in Canada, but also worldwide.
With the objective to gain insight in the atmospheric fate of α- and β-TBECH in general and in Southern
Canada specifically, we sought to characterise the spatial and seasonal variability in their air
concentrations and to explore their potential for atmospheric deposition. We also compare the
atmospheric fate of TBECH with that of the PBDEs, in particular 2,2',4,4'-tetrabromodiphenyl ether (BDE-
47), one of the traditional BFRs that has been internationally banned under the UNEP-SC-POPs but can
still be detected in the environment. Spatial patterns were investigated with two passive air sampler (PAS)
networks in the coastal regions of Quebec (QC) and British Columbia (BC). Seasonal trends in air
concentrations were studied using active air samplers (AAS) at an urban site in Toronto and two remote
sites on either coast. The input to aquatic systems was studied by measuring year-round wet deposition
and by taking passive water samples (PWS) at selected coastal sites.

## 2. Materials and methods

### 2.1 Sampling

#### 2.1.1 Passive air sampling network

Networks of XAD-2 resin-based PASs (Wania et al., 2003) were installed in QC along the ca. 1000-km
stretch of the St. Lawrence River and Estuary between Montreal and the Gulf of St. Lawrence, as well as



in BC on the coast of the Central Salish Sea. PAS housings, containing two XAD-resin filled mesh cylinders
each, were deployed at a height of ca. 3 to 5 m above ground on trees or other structures at 118 unique
sites between 2019 and 2022. Because PASs were deployed at some sites more than once, there was a
total of 169 PASs. Maps of the networks are shown in Figure S1, while coordinates of the sampling sites,
dates of deployment and retrieval, and deployment duration are provided in Table S1 in the Supporting
Information. By revealing the spatial variability in the air concentration of TBECH and other BFRs across a
wide region, the networks can identify the location of emissions to the atmosphere. The networks
included sites in highly urbanized and industrialised area, but also rural and remote sites. Additional
details are provided in Text S1. After retrieval, the PASs were stored in metal shipping containers sealed
with Teflon tape coated stoppers at -20°C until analysis.
2.1.2 Active air sampling and precipitation collection
To quantify the seasonal variability in the air concentration of TBECH and other BFRs in an urban source
region, 48 consecutive week-long AASs were taken on the campus of the University of Toronto
Scarborough in the eastern suburbs of Toronto, Ontario between June 2020 and May 2021 (Li et al., 2023).
These measurements were complemented by 24-hour AAS taken once a month for one year at two
remote coastal sites far from large urban centres and therefore, less likely to be influenced by local
emission sources. These were on Saturna Island, BC (L43; ca. 42 km NNE of Victoria; pop. ~300; Dec. 2019
- Nov. 2020; n=11) and Tadoussac, QC (S57; ca. 190 km NE of Quebec City; pop. ~800; Dec. 2020 and Nov.
2021; n=12). Contaminants in the particle and gas phase were collected on a glass-fiber filter (GFF) and a
polyurethane foam (PUF)/XAD-2/PUF sandwich, respectively. The sampling periods are listed in Table S2.
At Saturna Island and Tadoussac, monthly-integrated wet deposition samples were taken during the same
twelve-month period as the AAS. Precipitation samples were collected for one month each in sample
bottles containing 0.2 L dichloromethane, which were connected to overflow bottles to capture any
overflowing precipitation from the sample bottles. The sampling periods are listed in Table S3.
2.1.3 Passive water sampling network
Passive water samplers consisting of low-density polyethylene (LDPE) sheets in a metal mesh cage were
deployed in 20 unique locations in BC and QC during the late spring and summer of 2021 (Table S9). Figure
S2 provides maps with the sampling sites. Before deployment, the LPDE sheets were infused with several
performance reference compounds (PRCs) to determine sampling rates. Details are provided in Text S2.



## 2.2 Sample analysis

The gas phase sorbent from the AAS at Saturna Island and Tadoussac was Soxhlet extracted. All other air samples underwent pressurised liquid extraction using an accelerated solvent extractor. The precipitation samples were filtered through 0.7 µm GFFs and the filtered samples were then subjected to liquid-liquid extraction. The LDPE sheets were simply soaked in solvent overnight. Fourteen $^{13}$C-labelled surrogates were spiked into all samples prior to extraction. All extracts were reduced in volume and dried with $Na_2SO_4$. Injection standards ($^{13}$C-PCB-105 and 180) were added to the final extracts. Details on the extraction solvents, conditions, and the standards are provided in Text S3.

All samples were analyzed for a suite of 28 BFRs, including 15 PBDE congeners (BDE-17, 28, 47, 49, 66, 71, 85, 99, 100, 138, 153, 154, 183, 190 and 209), α-TBECH, β-TBECH, allyl 2,4,6-tribromophenyl ether (ATE), 2-bromoallyl 2,4,6-tribromophenyl ether (BATE), pentabromobenzene (PBBz), hexabromobenzene (HBBz), pentabromotoluene (PBT), pentabromoethyl benzene (PBEB), 2,3-dibromopropyl-2,4,6-tribromophenyl ether (DBTE), 2-ethylhexyl-2,3,4,5-tetrabromobenzoate (EHTBB), 1,2-bis(2,4,6-tribromophenoxy)ethane (BTBPE), bis(2-ethylhexyl)-3,4,5,6-tetrabromo-phthalate (BEHTBP), and decabromodiphenyl ethane (DBDPE). Four additional halogenated flame retardants (Dec-602, Dec-604, *syn*-DP, *anti*-DP) were also analyzed. Details of the instrumental analysis are provided in the Text S4. This is also where information on the QA/QC procedures can be found, which includes many procedural and field blanks. Limit of Detection (LOD) and Limit of Quantitation (LOQ) for different types of samples are provided in Table S12.

Both α- and β-TBECH are chiral molecules but can be expected to enter the environment as a racemate, i.e., with their two enantiomers being equally abundant (mean enantiomeric fraction $EF_{standards}$ = 0.502 ± 0.001). The unequal abundance of two enantiomers in environmental samples has been used to identify the occurrence of enantioselective processing, such as microbial transformation reactions. The enantiomeric composition of α- and β-TBECH was determined in samples with TBECH concentrations > LOQ from the PAS network in BC and QC, from the AAS in Toronto, and in a few PWS. The amounts of β-TBECH in the extract were generally too low for reliable chiral analysis, therefore, only results on the enantiomeric analysis of α-TBECH are presented. A description of the analytical method for enantiomeric analysis is provided in Text S4.



2.3 Air and water concentration calculations
Volumetric air concentrations were derived from the amounts quantified in the PAS sorbent using
sampling rates reported by Li et al. (2023). Volumetric water concentrations were calculated from the
amount quantified in the LDPE sheets using the dissipation of PRCs and the approach by Booij and Smedes
(2010). Details are again in Text S5. For statistical purposes, the measurements below the LOD are
represented by the value of the compound specific LOD (Table S12).
2.5 Partitioning properties calculations using COSMO-RS and other prediction tools
Equilibrium partition ratios between octanol and water ($K_{ow}$), octanol and air ($K_{oa}$), and air and water ($K_{aw}$)
of TBECH were estimated using quantitative structure-activity relationship (QSAR) models integrated in
several chemical property prediction tools (EASE-Suite, EPI Suite, OPERA). Because these QSARs cannot
distinguish between the properties of α- and β-TBECH, we also applied COSMOtherm, which is based on
quantum chemistry and statistical thermodynamics. Details of these approaches have been described
previously (Baskaran et al., 2021). More details on COSMOtherm are also given in Text S6. In general, the
predicted partition ratios of TBECH from the different models are in good agreement, i.e., are typically
within 0.30 log units of each other (Table 1). COSMOtherm tends to give slightly lower log $K_{OW}$ and log $K_{AW}$
values, suggesting a higher solvation in the aqueous phase than the QSARs. COSMOtherm-predicted
properties for the two isomers are also very similar, with α-TBECH being slightly more volatile than β-
TBECH.
# 3. Results
3.1 Air concentrations
3.1.1 Absolute concentrations ranges, isomer composition, and comparison with previous
measurements
α- and β-TBECH and BDE-47 were the most consistently detected BFRs in air samples taken across Canada,
detected in 69%, 59%, and 80% of samples from the PAS network, respectively. Table 2 summarises the
results, whereas data for each individual sample are documented in Tables S1 and S2. To a lesser extent,
several other EBFRs and PBDE congeners were also detected in the samples of this study, with their
detection frequencies or concentrations summarized in Tables S4 to S8. Because of their much higher
detection frequencies, the remainder of the manuscript is focused on TBECH and BDE-47.



To place these concentrations into context, Table S13 summarises all atmospheric concentrations of
TBECH previously reported in the literature. The concentrations of TBECH recorded here are at the lower
end of the range of levels detected in urban outdoor air elsewhere. In particular, levels in Canadian urban
locations are similar to those reported for outdoor air in Stockholm, Sweden (Newton et al., 2015; Wong
et al., 2018), but somewhat lower than those measured in Birmingham, UK (Drage et al., 2016), Brno,
Czech Republic (Bohlin et al., 2014; Melymuk et al., 2016) and previous measurements taken in Toronto
(Shoeib et al., 2014). Much higher values had been reported for indoor air (Cequier et al., 2014; Genisoglu
et al., 2019; Melymuk et al., 2016; Newton et al., 2015; Newton et al., 2016; Tao et al., 2016; Wong et al.,
2018), an electronic waste facility in China (Hong et al., 2018) and - somewhat incongruously - in
Longyearbyen, Svalbard (Carlsson et al., 2018). PAS deployed in remote regions (pop. <10 000 in a 20-km
radius) tended to have levels of TBECH below the LOD or LOQ. The levels above LOD measured in non-
urban locations in this study are among the lowest ever reported, comparable to what has been reported
for Tibet and Antarctica (Ma et al., 2017; Zhao et al., 2020). However, previous measurements of TBECH
in air from non-urban locations are rare. Overall, the alignment of the air concentration data with those
reported previously supports the validity of the results of this study.
BDE-47, despite its international ban under the Stockholm Convention, can still be detected in the
environment due to its persistence. Its presence in the atmosphere, along with other BDEs, has been
documented over the years in Canada, such as in Ottawa, ON (Wilford et al., 2004), Alert, NU (Wong et
al., 2021; Xiao et al., 2012), Yukon Territory (Yu et al., 2015), and the Great Lakes Basin (Shunthirasingham
et al., 2018). In this study, BDE-47 was also detected in the air in all sampling regions, with comparable
levels to TBECH. On Saturna Island, BDE-47 gas phase concentrations have decreased by one order of
magnitude relative to almost two decades prior (Noël et al., 2009). Levels in the atmospheric particle
phase and in precipitation in the area have similarly been decreasing.
3.1.2 Spatial variability of air concentrations in Canadian coastal regions
The spatial patterns in the air concentrations of $\alpha$- and $\beta$-TBECH in the coastal regions of BC and QC as
obtained from the PAS networks (Figure 1) show elevated levels in populated and urban areas. Specifically,
in QC, higher levels are observed along the St. Lawrence River corridor between Montreal and Quebec
City contrasting with lower levels on the shores of the St. Lawrence Estuary. In BC, higher levels are
apparent in the lower mainland and in Victoria. The overall higher levels observed in the PASs from BC
when compared to those from QC are likely the result of more urbanized sampling locations in the former.



Earlier studies also indicate that urban areas have higher TBECH air concentrations than remote regions
(Table S13).
The α-/β-TBECH ratio at sites where both isomers were present above their LODs (Figure S3) is remarkably
consistent in space on either coast, i.e., there is no indication that the ratio varies in space, e.g., by being
correlated with the absolute concentration level. The ratio was typically close to or above one at sites
with detectable amounts of both isomers in the air. An exception (α-/β-TBECH = 0.29) was at a site located
10 kilometres away from Quebec City, QC (S26). However, because the air concentrations of both isomers
measured at this site were below their LOQs, this value should be interpreted with caution.
Not only are the volumetric air concentrations of BDE-47 on the same order of magnitude as TBECH, but
they also share a similar spatial distribution in the atmosphere (Figure 1). This is also apparent from
significant correlations between the concentration of both isomers of TBECH and those of BDE-47 in the
PAS (Figure S4, Table S14, $R^2 > 0.24$, $p < 0.0001$), which seem to be strongly influenced by the urban sites
with elevated BFR concentrations. One difference is the notable presence of BDE-47 at one site in Alma,
QC (S48) during the first deployment period, despite BDE-47 at other sites in Alma (S49-53) and in the
wider Saguenay region (S54-56) being below the LOQ. PASs deployed at the same site (S48_2) and a site
in the vicinity (S54_2) a year later had BDE-47 levels below the LOQ, suggesting that the first data point at
S48 was an outlier and not an indication of a local point source.
In the BC PAS network, air concentrations are linearly correlated with population within a 20-km radius
around a PAS deployment site (NASA, 2015) (Figure S5, Table S15), more so for α- and β-TBECH ($R^2 = 0.27$
and 0.23, respectively) than for BDE-47 ($R^2 = 0.09$). The relationships were generally stronger when
explored separately with concentration data obtained at different average deployment temperature
(<10°C, 10-15°C, >15°C; Figure S5). Weaker or absent relationships at the warmest temperature (>15°C)
are likely caused by the relatively small number of summer deployments. Increasing slopes of these
relationships at warmer temperatures indicate a higher seasonal concentration amplitude at sites in
populated areas than in remote regions. This is also consistent with the expectation of a stronger
temperature dependence of the atmospheric concentrations of semi-volatile chemicals in source areas
than at sites without local sources (Wania et al., 1998). To further demonstrate this relationship with both
temperature and population, multiple linear regression was used on the log-transformed partial pressure
of TBECH against population and the inverse temperature, which resulted in higher correlations (Adjusted
$R^2 = 0.56$ and 0.42 for α- and β-TBECH, respectively). The seasonal variability in the air concentrations of
TBECH and BDE-47 is discussed in more depth when discussing the AAS results in section 3.1.3.



**Table 1**    Equilibrium partition ratios of α- and β-TBECH calculated by various physical-chemical property prediction tools at 0°C and 25°C.

| | | Log $K_{ow}$ | | Log $K_{aw}$ | | Log $K_{oa}$ | |
|---|---|---|---|---|---|---|---|
| | | 0°C | 25°C | 0°C | 25°C | 0°C | 25°C |
| COSMOtherm | α-TBECH | 4.49 | 4.45 | -4.67 | -3.82 | 8.93 | 8.36 |
| | β-TBECH | 4.39 | 4.35 | -4.92 | -4.07 | 9.08 | 8.51 |
| EAS-E Suite | TBECH | 5.70 | 5.43 ± 0.52 | -4.41 | -3.34 ± 0.36 | 10.14 | 8.80 ± 0.81 |
| EPI Suite | | | 5.24 | | -2.77 | | 8.00 |
| OPERA | | | 5.24 ± 0.03 | | -3.72 ± 1.20 | | 8.42 ± 0.13 |

**Table 2**    Detection frequency, median, mean, and maximum of the concentrations and wet deposition fluxes of the three most frequently detected BFRs in air samples (gas phase) and water samples (dissolved) in Canada. For statistical purposes, measurements below the LOD were represented by the value of the compound specific LOD.

| | n | -TBECH | | | | -TBECH | | | | BDE-47 | | | |
|---|---|---|---|---|---|---|---|---|---|---|---|---|---|
| | | n > LOD | med. | mean | max | n > LOD | med. | mean | max | n > LOD | med. | mean | max |
| **Concentrations in Passive Air Samples in pg/m³** | | | | | | | | | | | | | |
| QC | 86 | 39 | <LOD | 0.22 | 1.21 | 30 | <LOD | 0.22 | 0.99 | 60 | 0.84 | 1.0 | 6.1 |
| BC | 83 | 77 | 0.44 | 0.54 | 2.20 | 70 | 0.38 | 0.46 | 2.1 | 76 | 1.5 | 1.9 | 10 |
| **Concentrations in Active Air Samples in pg/m³** | | | | | | | | | | | | | |
| Tadoussac | 12 | 10 | 0.09 | 0.13 | 0.29 | 9 | 0.06 | 0.08 | 0.17 | 11 | 0.13 | 0.23 | 0.69 |
| Saturna | 11 | 8 | 0.06 | 0.07 | 0.13 | 6 | 0.04 | 0.03 | 0.12 | 11 | 0.57 | 0.55 | 1.1 |
| Toronto | 48 | 48 | 0.30 | 0.40 | 1.34 | 48 | 0.20 | 0.26 | 0.87 | 48 | 0.66 | 1.5 | 9.3 |
| **Concentrations in Passive Water Samples in pg/L** | | | | | | | | | | | | | |
| QC | 12 | 0 | <LOD | <LOD | <LOD | 0 | <LOD | <LOD | <LOD | 12 | 1.5 | 2.0 | 6.2 |
| BC | 36 | 25 | 0.19 | 0.40 | 2.2 | 19 | 0.10 | 0.26 | 1.4 | 36 | 1.2 | 1.4 | 3.8 |
| **Concentrations in Precipitation Samples in pg/L** | | | | | | | | | | | | | |
| Tadoussac | 11 | 11 | 67 | 103 | 302 | 11 | 49 | 83 | 251 | 11 | 32 | 42 | 90 |
| Saturna | 12 | 12 | 233 | 464 | 1416 | 12 | 183 | 377 | 1125 | 11 | 23 | 36 | 102 |
| **Wet deposition fluxes in pg/m²/day** | | | | | | | | | | | | | |
| Tadoussac | 11 | 11 | 96 | 154 | 628 | 11 | 82 | 124 | 487 | 11 | 66 | 71 | 153 |
| Saturna | 12 | 12 | 697 | 759 | 2099 | 12 | 574 | 622 | 1731 | 11 | 53 | 127 | 468 |



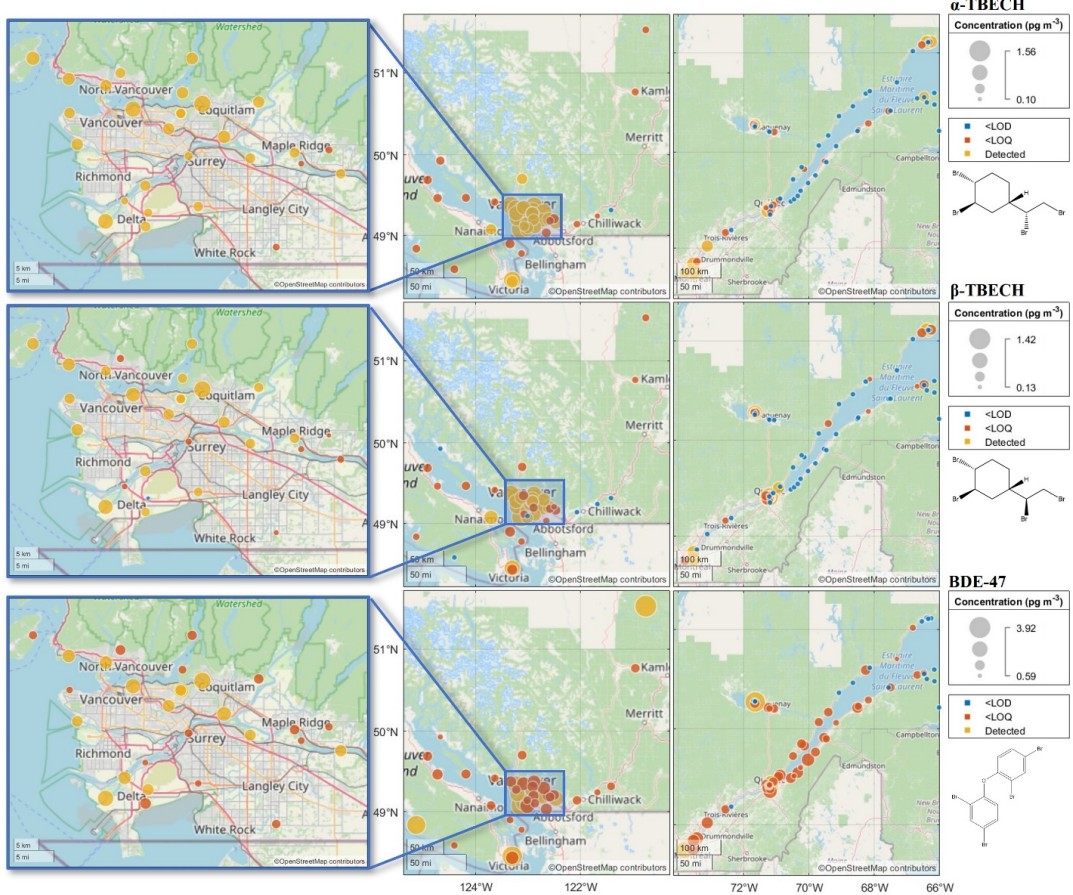

254

Fig. 1    The spatial distribution of the air concentrations of α-TBECH (top), β-TBECH (middle), and BDE-47 (bottom) in British Columbia and Quebec. Close-up maps (left) provide a detailed view of the Vancouver metropolitan area. Average air concentrations are displayed for sites with replicate deployments; therefore, the concentrations indicated on the figure may not align with concentrations listed in Table 2 and Table S1. Air concentrations below the LOD are marked in blue and display the value of the LOD. Air concentrations above the LOD and below the LOQ are marked in dark orange. Air concentrations above LOQ are marked in yellow. Concentration values are listed in Table S1.

### 3.1.3 Seasonal variability of air concentrations in urban and remote regions of Canada

*Seasonal variability in Toronto*

The seasonal trends of AAS-derived air concentrations of TBECH and BDE-47 in Toronto, ON (Table S2, Figure 2A and 2B) indicate a strong relationship with average air temperature (Figure 2C), being higher in the summer (April to August) and lower in winter (September to March). Accordingly, log-transformed partial pressures ln(p/Pa) of α- and β-TBECH and BDE-47 were significantly correlated with inverse



absolute temperature (Figure S6, Table S16, $R^2$=0.61, 0.55, and 0.81, respectively, $p<0.0001$). Such a
temperature dependence of the atmospheric concentration of semi-volatile organic compounds is often
interpreted as temperature driven air-surface exchange (Wania et al., 1998) or temperature-dependent
rates of emission.

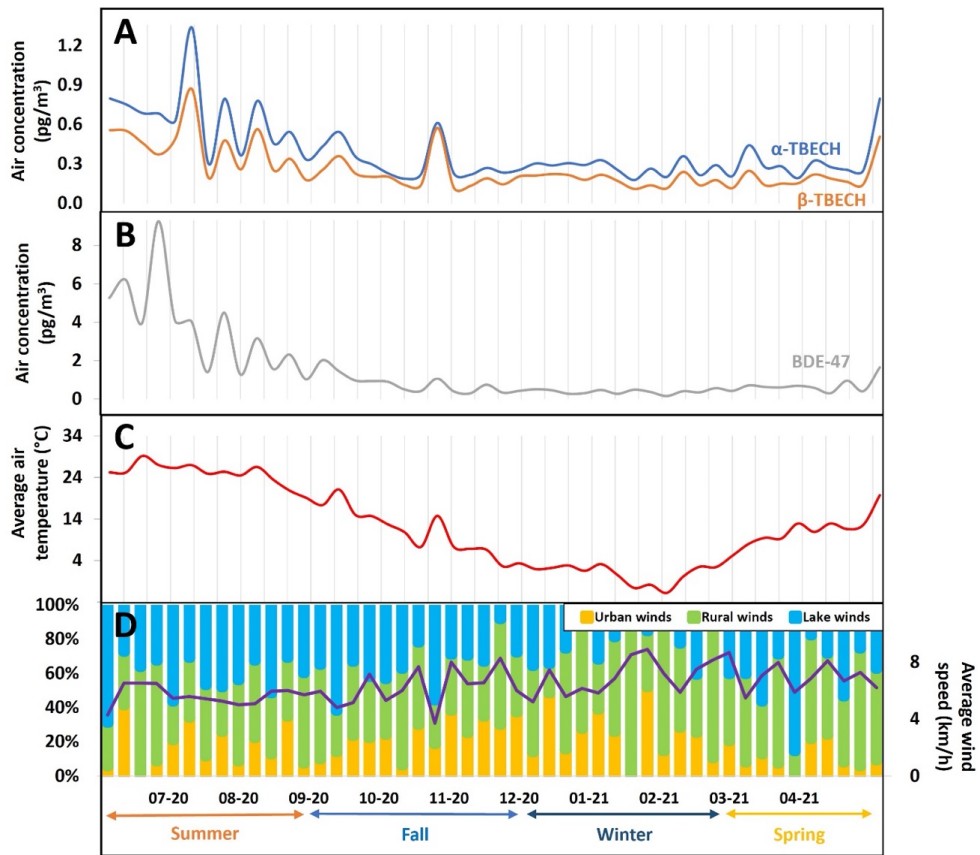


Fig. 2    The air concentration of α-TBECH, β-TBECH, and BDE-47 and meteorological parameters in
         Toronto, Ontario during 48 weekly active air sampling periods between June 2020 and May
         2021.

With the sampling site located in an Eastern suburb of a large urban conglomeration, wind also influenced
the air concentrations of TBECH and BDE-47. Higher levels were recorded when weak winds were blowing
from the urban core of Toronto, approximately 20 km to the SW of the sampling site. Text S7 and Figure
S7 provide detail on how an urban wind fraction was derived. The average wind speed and the fraction of
wind originating over urban areas, rural regions and Lake Ontario are shown in Fig. 2D. Multivariate
regressions of the ln(p/Pa) with both reciprocal temperature and the urban wind fraction are highly



significant for all three BFRs and stronger than univariate regressions with reciprocal temperature only
(Table S17).
*Seasonal variability in Saturna Island and in Tadoussac*
The AAS-derived air concentrations of TBECH and BDE-47 on Saturna Island were very low (Table S2).
Whereas BDE-47 showed a clear seasonal pattern with higher levels in summer than in winter, any
seasonal pattern of the TBECH isomers was obscured by levels below the LOD in air samples taken in
December, June and July (Figure S8). Accordingly, whereas the ln(p/Pa) of BDE-47 was significantly
correlated with reciprocal temperature ($R^2$=0.44 and *p*=0.027), for both TBECH isomers those
relationships were generally weak and insignificant (Table S16). Weak or absent ln(p/Pa) versus 1/T
relationships are typical for remote sites without local sources, where the concentration of a compound
is controlled by advection from elsewhere (Wania et al., 1998).
In contrast to Saturna Island, AAS-derived air concentrations of TBECH and BDE-47 in Tadoussac (Table
S2) showed a clear seasonal trend with higher levels in summer and levels often below the LOD in winter
(Figure S9). Ln(p/Pa) versus 1/T regressions (Figure S6) were significant but only explained a relatively
small fraction of the overall variability ($R^2 > 0.30$; Table S16). One factor that may have contributed to the
differences in the seasonal patterns of TBECH at the two coastal sites are the much larger seasonal
temperature range in QC (-12 to +18 °C) than in BC (+1 to +18 °C).
We also sought to relate the temporal variabilities in the measured air concentrations with the history
and origin of the sampled air masses using the Lagrangian atmospheric dispersion model FLEXPART (Pisso
et al., 2019). TBECH levels below the LODs in June and July on Saturna Island despite the relatively warm
temperatures may be related to air masses arriving at the sampling site directly from the Pacific Ocean,
i.e., without encountering any potential urban source areas. Overall, TBECH levels on Saturna Island arise
from competing influences of air mass origin and temperature. More detail in provided in Text S8.
3.2 Concentrations in passive water samples
The water concentrations of the BFRs measured with PWSs in the coastal regions of BC and QC are
summarized in Table 2 with all data provided in Table S9. Whereas the TBECH isomers were below the
LODs in all water samples from QC, in BC, α- and β-TBECH ranged between <LOD to 1.75 pg/L and <LOD
to 1.15 pg/L, respectively, and were typically above the LODs at sites close to urban centers (Victoria,
Vancouver) (Figure S10). BDE-47 was detected at all sites of this study, with higher levels close to Victoria



(Figure S12). BDE-47 water concentrations were also generally slightly higher compared to those of
TBECH, albeit on the same order of magnitude.
We compared the results from the PWS network in this study with water concentrations reported by the
Federal Whales Initiative (FWI) of Canada (ECCC, 2022). Sampling campaigns conducted by the FWI
occurred in the Fraser River, BC, including its main tributaries (Thompson River and Harrison River)
between 2019 to 2021. The median water grab sampling concentrations of TBECH (sum of all isomers)
and BDE-47 reported by the FWI were 22.9 pg/L (presumably incorrectly labelled as ng/L in the database)
and 32 pg/L, respectively, which are one to two orders of magnitude higher than the PWS measurements.
However, the median calculated for TBECH only reflects the three out of 122 samples analyzed in total
with concentrations above the LOD of 1.22 pg/L. This may be attributed to the grab sampling technique
used and the FWI sampling at inland freshwater sites, whereas the PWSs in this study were deployed in
sea water, which is expected to be more diluted. BDE-47 concentrations measured in the water of the
Juan de Fuca Strait, BC (Sun et al., 2023) were at comparable levels (0.6-2.0 pg/L) to the ones in this study,
giving credence to this theory.
3.3 Concentrations in precipitation samples
TBECH and BDE-47 could be quantified in almost all precipitation samples from Saturna Island and
Tadoussac (Table S3), with levels generally being higher on the west coast. The concentrations of TBECH
varied strongly between different months, without displaying a clear seasonal trend (Figure S13). Both
isomers exhibited similar fluctuations. The BDE-47 concentrations in precipitation were generally lower
and did not fluctuate as strongly as TBECH.
3.4 Enantiomeric fractions of α-TBECH
The results of the chiral analysis are presented using enantiomeric fractions (EFs). A racemate with equal
amounts of the (+) and (-) enantiomers has an EF of 0.50. EFs that are significantly above or below 0.5
indicate the occurrence of enantioselective processes, typically biological pathways relying on enzymes
or enantioselective membranes. The α-TBECH standard had an EF = 0.502± 0.001. The EFs of α-TBECH in
the PAS extracts had values that were significantly above and below this value, ranging from 0.32 at L2 to
0.66 at L6 (Table S18). While PAS extracts from non-urban sites rarely had sufficient amounts of α-TBECH
for reliable chiral analysis, samples from a few coastal sites on the St. Lawrence Estuary and in the
Saguenay area in QC and on Vancouver Island in BC had EFs below 0.50 (Figure 3). Conversely, samples
from populated urban sites in Quebec City and Vancouver generally had EF greater than 0.50. Several sites




in the Vancouver area and in Victoria with EF < 0.50 are located on the shore of Burrard Inlet/Vancouver
Harbour and Oak Bay, respectively.

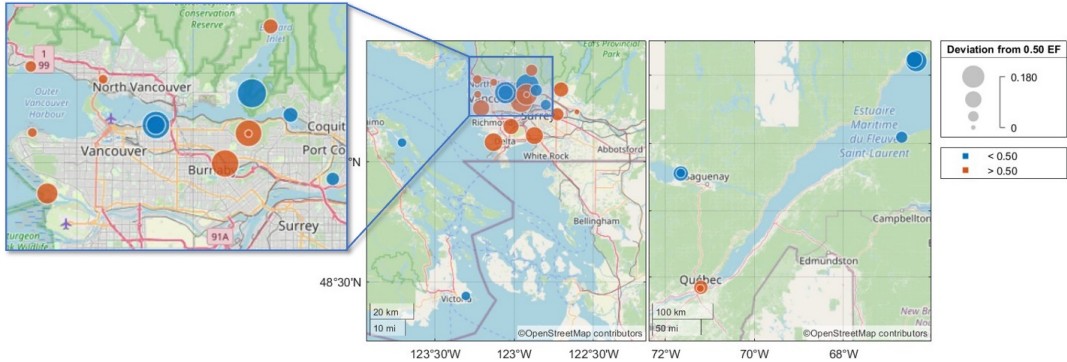


Fig. 3    The spatial distribution of the enantiomeric fraction (EF) of α-TBECH in selected BC and QC PAS.
The absolute deviation from 0.50 (i.e., racemic mixture) of each site is plotted on the map, with
values further from 0.00 indicating higher enantiomeric excess of one enantiomer. Sites with EF
< 0.50 and EF > 0.50 are marked in blue and red, respectively.

Consistent with the PAS from urban deployments in BC and QC generally having EFs > 0.5, the EFs in the
AAS from Toronto were also mostly above 0.502±0.001 (only one out of 36 samples had an EF<0.502)
(Table S19). While there was no clear seasonal trend in the EF, α-TBECH tended to be close to racemic in
summertime air (Figure S14, Table S20; $R^2$=0.14, $p$=0.023). Conversely, the EFs of α-TBECH in the PWS
extracts (Table S21) were below 0.5, agreeing with the EF spatial trends observed in the PAS network.
Despite its urban origin, the TBECH in Canada has clearly experienced enantioselective processing, i.e.,
has evaporated from reservoirs where it has undergone microbial transformations. This is in agreement
with the observed enantiomeric excess for α-TBECH in soil experiments, with increasing excess seen after
prolonged degradation (Wong et al., 2012). The consistent EF < 0.5 observed in water and marine air
samples also suggests that at least some of the TBECH in the atmosphere evaporated from seawater.
Moreover, the microbial processes occurring in urban soils and in marine waters seem to favour opposite
enantiomers, leading to the divergent enantiomeric enrichment of TBECH.



## 4. Discussion

### 4.1 Atmospheric deposition of TBECH

The log $K_{OA}$ values less than 9 (Table 1) imply that TBECH will not sorb appreciably to atmospheric particles. This was confirmed by the failure to detect either isomer in the GFFs of the AAS in Saturna and Tadoussac. Atmospheric deposition thus must occur by diffusive gas exchange or by precipitation scavenging from the gas phase. We paired the analytical results in different types of samples to study these deposition processes.

#### 4.1.1 Diffusive air-water gas exchange

By having measured TBECH and BDE-47 in both air and water, we can investigate the water-air equilibrium status and the potential direction of diffusive air-water gas exchange in the Salish Sea. We estimated water-air fugacity ratios $f_W/f_A$ from the measured concentrations as described in Text S9. The $f_W/f_A$ ratios for both TBECH isomers and BDE-47 were below unity throughout BC, indicating net deposition of the BFRs from the atmosphere to the Salish Sea. This is also the case for Vancouver and Victoria Harbour (Figure S15, Figure S16). While this analysis could not be done in QC for TBECH, as it could not be detected in water, the $f_W/f_A$ ratios for BDE-47 in QC were found to also be below unity (Figure S16). We caution that these air-water exchange calculations have considerable uncertainty, which stems from the uncertainty (i) of volumetric air and water concentrations derived from passive sampling techniques, (ii) in the $K_{AW}$, which is apparent from differences in the predictions of different methods (Table 1), and (iii) arising from combining air and water data obtained during different time periods. While periods of deployment overlapped in this study, PWS were deployed for approx. one month (Table S9), whereas PAS deployment periods were considerably longer (Table S1).

#### 4.1.2 Scavenging ratios

The monthly measurements of BFRs in air and precipitation from Saturna Island and Tadoussac were used to estimate scavenging ratios, i.e., the concentration of TBECH and BDE-47 in precipitation divided by their concentration in air (Text S9). The scavenging ratios of the TBECH and BDE-47 ranged between $10^5$ to $10^7$, and $10^4$ to $10^5$, respectively (Figure S17). Exchange of vapours between the air and water droplets in the atmosphere is assumed to occur fast enough to achieve almost instant equilibrium. If equilibrium between the precipitation droplets and atmospheric gas phase was established, the scavenging ratio should equal the water-air partition ratio, $K_{WA}$ (=$1/K_{AW}$). The measured scavenging ratios for TBECH in this study were one to three orders of magnitude higher than the predicted $K_{WA}$ values (Table 1). Accordingly, rainwater-



air fugacity ratios of the BFRs in Saturna Island and Tadoussac (Figure S18) were one to three orders of
magnitude greater than the expected equilibrium value, i.e., one.
Several reasons may contribute to this deviation from equilibrium. We have combined air concentration
measurements for a 24-hour period with precipitation samples collected over the course of an entire
month. The AAS quantification is uncertain, due to the very low levels in the AAS extracts, as is the $K_{AW}$
and the temperature. Other potential reasons include that adsorption occurs at the water-air interface
(Hoff et al., 1993), or that the BFRs experience sorption to dissolved organic matter and other sorption
phases (Poster & Baker, 1996), all which could lead to higher scavenging ratios. However, information on
interface adsorption coefficients of the BFRs and other sorption phases at Saturna Island and Tadoussac
is limited.
Seasonal differences in scavenging ratios for some organic contaminants have been observed previously,
with higher values measured during colder seasons (Wania & Haugen, 1999). This is likely because $K_{WA}$
increases as the temperature lowers, resulting in more efficient scavenging. Another hypothesized reason
is that snow at equivalent water content, having greater surface area, can act as a better scavenging
medium than rain (Lei & Wania, 2004; Paramonov et al., 2011). The logarithm of the inverse scavenging
ratio, -ln(SR) (i.e., ln(1/SR)), was regressed with the reciprocal of the average air temperature. Whereas
the scavenging ratios of TBECH did not significantly correlate with temperature (Table S22), the
scavenging of BDE-47 at both sites increased with decreasing temperature (Figure S19).
4.2 Do the two TBECH isomer show divergent atmospheric fate?
In this study, the relative abundance of α- and β-TBECH was very consistent, with mean α-/β-TBECH ratio
generally ranging from 1.1 to 1.6, i.e., indicating a slightly higher abundance of the α-isomer relative to
the β-isomer. In particular, this ratio was very similar in different parts of the country, with median α-/β-
TBECH in active air samples with concentrations of both isomers above the LOD being 1.56 (range 1.07 to
2.01) in Toronto, 1.72 (range 1.44 to 3.28) in Tadoussac, and 1.57 (range 1.08 to 2.28) on Saturna Island.
Similarly, the median α-/β-TBECH in PAS was the same in QC (1.12, range 0.29 to 1.99) and BC (1.16, range
0.51 to 3.20). This slightly higher abundance of the α-isomer relative to the β-isomer is consistent with
earlier reports of TBECH elsewhere in the atmosphere (Table S13).
The relative abundance of the two isomers is also generally very similar in different environmental media.
The mean α-/β-TBECH ratio in AAS (QC mean = 1.95; BC mean = 1.62; ON mean = 1.56) and PAS (QC mean
= 1.03; BC mean =1.21) bracket the values recorded in the PWS (BC mean = 1.52) and precipitation (QC





mean = 1.21; BC mean=1.22) (Figure 4). Moreover, isomer abundance was very similar in different samples
of the same type. The two isomers exhibited nearly identical spatial patterns in air (Figure S3) and water
(Figure S11). The seasonal patterns in air (Figure 2, Figure S8, Figure S9) and in precipitation (Figure S13)
were also highly consistent for α- and β-TBECH. Not surprisingly, α- and β-TBECH had similar water-air
equilibrium status in the Salish Sea (Figure S15) and essentially identical scavenging ratios (Figure S16).

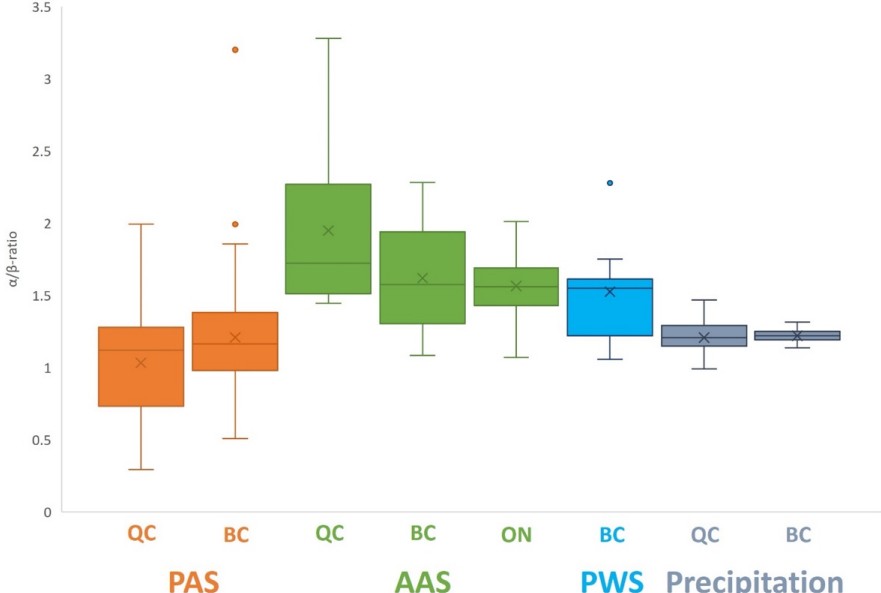


Fig. 4    Comparison of the α-/β-TBECH ratio calculated in the sampling regions of this study, grouped in
different environmental media/sampling methods. The mean α-/β-TBECH ratio is indicated with
an "x" for each method. For statistical analysis, outliers (α/β > 2.0) were included but are explicitly
marked on the plot as dots. Only measurements with both isomers above the LOD were used in
this plot.

Taken together, this consistency in TBECH composition suggests that α- and β-TBECH exhibit similar
atmospheric behaviours. This is also consistent with the very minor differences in partitioning behaviour
predicted for the two congeners by COSMOtherm (Table 1). The slightly lower COSMOtherm-predicted
air-water and slightly higher octanol-air partition ratios of the β-isomer compared to the α-isomer are not
apparent in divergent atmospheric behaviour.





### 4.3 Sources of TBECH to the Canadian atmosphere

While literature on TBECH in the atmosphere has been limited, the few existing studies all point to TBECH levels generally being higher in populated urban areas. Using the spatial concentration patterns from this study, the first to compare urban and remote concentrations of TBECH within a single study, we were able to confirm that TBECH strongly correlates with population. Moreover, the temporal trends of TBECH indicate that the air concentrations increase during the warmer season with fixed isomeric composition, particularly in the urban centres of Canada. Its year-round presence in remote regions in this study also suggests that TBECH has the potential for long range transport (LRT).

The inventory update phase 3 survey in 2017 under Canada's Chemical Management Plan (CMP) resulted in no reports of manufacturing or importing TBECH into the country by domestic companies, eliminating the possibility of point sources located in Canada for this flame retardant, such as factories that produce or import commercial TBECH. However, several recent international patents have included commercial TBECH as one of the flame retardants used mostly in electronics (Geng et al., 2018; Ke & Lv, 2020), pointing to the possibility that TBECH enters Canada almost exclusively as part of imported products. It is likely that most, if not all the TBECH detected in this study originate from the gradual release from these imported products. Populated urban areas, with typically greater overall usage of products containing flame retardants, e.g., electronics and plastics, have consistently shown higher TBECH levels with more pronounced temperature dependence compared to remote regions, and therefore suggest greater leakage. This is also consistent with previous studies reporting higher TBECH levels in indoor air, compared to outdoor (Table S13).

Chiral compounds are typically commercially produced as a racemate, including technical mixtures of TBECH (Ruan et al., 2018). Abiotic processes occurring in the environment, such as hydrolysis, oxidation, and volatilization, are unable to distinguish between enantiomers, and therefore, the EF of the compound will remain unchanged through these forms of physical processes. Microbial degradation, however, can result in the preferential consumption of one enantiomer over another, or in other words, deviations in the EF from 0.5. The variations in the EF of a compound can therefore be used to observe the enantiomeric transformations of a molecule in the environment. Therefore, the EF results of this study also indicate that secondary emissions of TBECH may occur in Canada. Once TBECH is deposited, a portion of it may undergo enantioselective processes in the environment, e.g., uptake, translocation, and metabolism by microbes. TBECH that re-volatilizes from these reservoirs has an excess in one enantiomer, with different enantiomer preferences seen in different environments (urban soils vs. marine waters).



## 5. Conclusion

Environmental assessments of chemicals are currently prioritized by the Government of Canada based on their persistence, bioaccumulation potential, and the quantity of domestic industrial manufacture, import, and/or export of a chemical. With neither official records of importation nor manufacturing existing in Canada, TBECH is, for environmental assessment purposes, not considered to be present in Canada. Therefore, despite being classified as persistent and bioaccumulative, it has been deemed a nonpriority for evaluation by Environment and Climate Change Canada (ECCC) and other governmental programs. However, the results of this study reveal that TBECH is ubiquitous in the Canadian atmosphere and waters at comparable levels with a legacy flame retardant, BDE-47, with elevated levels in populated areas. This, along with the strong seasonal variability of TBECH observed in urban areas, suggest that TBECH is emitted primarily from imported products. At least some of the TBECH in the atmosphere has been subject to microbial processing, i.e., entered the atmosphere after having been previously deposited to the surface (water/soil), resulting in enantiomeric enrichment.

With the current assessment requisites in place, TBECH and other chemicals of emerging concern (CECs) that are supposedly "nonexistent" in Canada would remain low priority for environmental assessment. This is particularly concerning for compounds that can enter in the country as part of imported products, potentially evading documentation. Such evasion has already been observed in Canada with short-chained chlorinated paraffins, which have been detected in imported products despite a ban on manufacture, usage, and import in 2013 (Kutarna et al., 2023). This raises the question whether the current assessment criteria are sufficient to address the relevant CECs in Canada. The increasing development and production of novel chemicals aimed to replace legacy POPs in their functions will only further highlight the limitations of the assessment requisites in place. We emphasize, with the results of this study, the importance of long-term environmental monitoring for EBFRs and other CECs as part of working towards proper screening and risk assessment.

## 6. Acknowledgements

We thank Geri Crooks, Alexandre Costa, Yannick Lapointe, Louis-Georges Esquilat, Jocelyn Praud, Sandrine Vigneron, François Gagnon, Jonathan Pritchard, Alessia Colussi, Christian Boutot, Bruno Cayouette, Fella Moualek, Frédérik Bélanger, Claude Lapierre, Félix Ledoux, Samuel Turgeon, Sarah Duquette and the CAPMON team for their assistance in deploying samplers and providing facilities/permissions to the sampling locations.



Financial support from Grant and Contribution Agreements (GCXE20S008, GCXE20S010, GCXE20S011)
with Environment and Climate Change Canada is gratefully acknowledged.

## 7. Data availability

All data generated in this study are provided in the supplement.

## 8. Author Contribution

FZ, YL, and JO prepared and extracted the passive air samplers (PASs) and the Toronto active air samples
(AAS). FZ and YL also took the Toronto AAS. YDL prepared standards and performed the instrumental
analysis on all samples. CS prepared and obtained samples from Saturna Island and Tadoussac as well as
the passive water samplers (PWSs) and performed the chiral analysis. KL and FAPCG deployed/retrieved
PASs and PWSs in BC. ABC, ADC, ZL, NA, HH, FZ and FW deployed/retrieved PASs and PWSs in Quebec.
provided guidance on sampling and sample analyses. JO compiled and interpreted data, with input from
FW. JO and FW wrote the manuscript with input by the other co-authors. HH coordinated the project.

## 9. Competing Interests

The authors have no competing interests to declare.

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
