# Peer review of "The atmospheric fate of 1,2-dibromo-4-(1,2-dibromoethyl)cyclohexane (TBECH)"

_EGUsphere, 2023_

## Community Comment (CC1)

**Review of: Oh, Jenny, The atmospheric fate of 1,2-Dibromo-4-(1,2-dibromoethyl)cyclohexane (TBECH…) (egusphere-2023-1151)**

This is an outstanding paper that examines in detail the concentrations and phase distribution of TBECH in air and water of the Canadian Environment. The sampling coverage in QC, BC and Toronto is very thorough, a lot of time went into designing and implementing the sampling networks and performing the subsequent analysis. The presentation of results and discussion of pathways and implications is also thorough and informative.

Here we have a flame retardant that is not used in Canada, but is nevertheless abundant in the environment, and must therefore have arrive by air transport. Enantiomeric analysis reveals that the TBECH has been subjected to degradation processes that deplete one enantiomer, thus pointing to "secondary sources". An unanswered question is from where?  My guess is the U.S., and a comment has been made about this near the end of the review.

Below are some points that could use a bit of clarification.  Overall, I support publication with minor revisions.

Line 54.  Would be good to state the percentages of the four TBECH isomers in the technical mix. Relevant also for Section 4.1.2.

Line 217. How do the ratios in air compare with those in the technical mix, and in other locations where TBECH in air has been reported?

Looking for Saturna Island and Tadoussac on the maps, but can't find them.  Possible to label the appropriate dots on the map?

Line 326. Section 3.3. Interesting that TBECH was prominent in Saturna Island precipitation, whereas concentrations in the AAS were quite low. The dimensionless scavenging ratios, SR = Cprecip/Cair, are on the order of $10^6$ - $10^7$ (Fig. S17). As the authors note in Section 4.1.2, such SRs are much higher than commonly encountered for gaseous chemicals and they are even on the high end of SRs for particle-bound chemicals. The authors discuss possible reasons for this in Section 4.1.2.  Not mentioned is the possibility that a differences in air masses might account for this.  On page S31 the authors state: "In the studied region, summer air comes from the Pacific Ocean and winter air from the continent."  Could there also be a vertical influence at this coastal site? Warm continental air with higher TBECH at the cloud level (scavenged by precip.) overriding cooler clean ocean air underneath (sampled by AAS)?

Line 332, Section 3.4.  The enantiomer elution order is not specified here.  In Supplementary Text 4, this statement is made: "To determine enantiomer fraction (EFs) for α-TBECH, the elution order was used and calculated as E1/(E1+E2), since the correspondence between optical signs and chromatographic elution is unknown for the TBECH (Wong et al., 2012)."  Could the authors also state that EF = E1/(E1+E2) in Section 3.4?

Line 409, Section 4.2. The title of the paper says "…atmospheric fate"…, and "fate" is partly accounted for through discussions of deposition and air-water exchange.  However, degradative fate is not discussed, and this section would be a good place for it.  Can some simple statements be made about predictive OH radical degradation (AOPWIN)?  My guess is that the model will not differentiate among the TBECH isomers, but at least would provide information to balance against deposition removal.

Line 437, Section 4.3.

It seems that the "elephant in the room" has been ignored in this discussion – the possibility of air transport from the U.S.! Especially since the sampling sites are not far from the border. What is known about TBECH use south of the border?

---

## Author Comment (AC1)

**Response to Terry Bidleman's community comment on: Oh, Jenny, The atmospheric fate of 1,2-Dibromo-4-(1,2-dibromoethyl)cyclohexane (TBECH…) (egusphere-2023-1151)**

Line 54. Would be good to state the percentages of the four TBECH isomers in the technical mix. Relevant also for Section 4.1.2.

Thank you very much for the suggestion. The manuscript will be modified to include the percentages of the isomers in the introduction. Changes are in bold: "While commercial TBECH comprises mostly the α- and β-isomers **(57.3% and 42.5%, respectively)** (Ruan et al., 2018), TBECH has been reported to thermally isomerize to the γ and δ isomers when heated to 123 °C (Arsenault et al., 2008)."

Line 217. How do the ratios in air compare with those in the technical mix, and in other locations where TBECH in air has been reported?

The comparisons between the ratios reported in our study and elsewhere is summarized in Table S13. This is further explored in section 4.2, when discussing the (lack of) divergent atmospheric fate of the isomers. However, based on the suggestion, the manuscript will be modified to explicitly state the comparison to reported ratios of the technical mixtures of TBECH in section 4.2: "This slightly higher abundance of the α-isomer relative to the β-isomer is consistent with earlier reports of TBECH elsewhere in the atmosphere (Table S13), **as well as with the reported abundances in technical mixtures (Arsenault et al., 2008; Ruan et al., 2018).**"

Looking for Saturna Island and Tadoussac on the maps, but can't find them. Possible to label the appropriate dots on the map?

The sites in Saturna Island and Tadoussac were mentioned in section 2.1.2 (L43 and S57, respectively). These sites can be located on Figure S1. This section will be updated to clarify: "These were on Saturna Island, BC (L43 **on Figure S1**; ca. 42 km NNE of Victoria; pop. ~300; Dec. 2019 - Nov. 2020; n=11) and Tadoussac, QC (near S57 **on Figure S1**; ca. 190 km NE of Quebec City; pop. ~800; Dec. 2020 and Nov. 2021; n=12)."

Line 326. Section 3.3. Interesting that TBECH was prominent in Saturna Island precipitation, whereas concentrations in the AAS were quite low. The dimensionless scavenging ratios, SR = Cprecip/Cair, are on the order of $10^6$ - $10^7$ (Fig. S17). As the authors note in Section 4.1.2, such SRs are much higher than commonly encountered for gaseous chemicals and they are even on the high end of SRs for particle-bound chemicals. The authors discuss possible reasons for this in Section 4.1.2. Not mentioned is the possibility that a difference in air masses might account for this. On page S31 the authors state: "In the studied region, summer air comes from the Pacific Ocean and winter air from the continent." Could there also be a vertical influence at this coastal site? Warm continental air with higher TBECH at the cloud level (scavenged by precip.) overriding cooler clean ocean air underneath (sampled by AAS)?

Differences in air masses is another plausible explanation that may contribute to the high scavenging ratios observed at Saturna Island. We will update section 4.1.2 to include this reasoning: "Other potential reasons include: **i) differences in air masses at different altitudes, where the warm continental air containing higher TBECH levels at the cloud level is scavenged by precipitation and has overridden the cooler clean ocean air at lower altitudes that was sampled by AAS**, ii) adsorption that occurs at the water-air interface (Hoff et al., 1993), or iii) the BFRs experience sorption to dissolved organic matter and other sorption phases (Poster and Baker, 1996), all which could lead to higher scavenging ratios. However, information on interface adsorption coefficients of the BFRs and other sorption phases at Saturna Island and Tadoussac is limited."

Line 332, Section 3.4. The enantiomer elution order is not specified here. In Supplementary Text 4, this statement is made: "To determine enantiomer fraction (EFs) for α-TBECH, the elution order was used and calculated as E1/(E1+E2), since the correspondence between optical signs and chromatographic elution is unknown for the TBECH (Wong et al., 2012)." Could the authors also state that EF = E1/(E1+E2) in Section 3.4?

Thank you very much for the suggestion. We will update the manuscript to include the EF formula in section 3.4: "The results of the chiral analysis are presented using enantiomeric fractions (EFs), **where the chromatographic elution order of the enantiomers was used to calculate EF=E1/(E1+E2).**"

Line 409, Section 4.2. The title of the paper says "…atmospheric fate"…, and "fate" is partly accounted for through discussions of deposition and air-water exchange. However, degradative fate is not discussed, and this section would be a good place for it. Can some

simple statements be made about predictive OH radical degradation (AOPWIN)? My guess is that the model will not differentiate among the TBECH isomers, but at least would provide information to balance against deposition removal.

We are grateful for being alerted to this omission and we will include a paragraph in section 4.2 that discusses what is known about the degradation reactions of TBECH in the atmosphere. The predicted rate constants will also be tabulated in Table 1:

**Information on the atmospheric degradation of TBECH has been extremely limited, with no experimental results in the literature to date. Using density functionals, Wang et al. (2021) predicted rate constants for the stereospecific hydroxyl radical-initiated transformation of 7.1 x $10^{-11}$ and 1.2 x $10^{-10}$ cm$^3$/s at 293 K for α- and β-TBECH, respectively, which, when applying 9.7 x $10^{-5}$ molecules/cm$^3$ as the concentration of OH radicals in the atmosphere, corresponds to atmospheric lifetimes of 4.0 and 2.4 hours. Such relatively short gas-phase lifetimes are difficult to reconcile with the relatively wide dispersal of TBECH recorded in the current study. Neither do our measurements support the idea that the atmospheric lifetime of the β-isomer is considerably short than that of *α*-TBECH, as this should have been apparent in an increasing relative abundance of *α*-TBECH with increasing distance from sources.**

**Rate constants predicted with COSMOtherm and the Atmospheric Oxidation Program for Microsoft Windows (AOPWIN) via EPI Suite (Table 1) are on the same order of magnitude and correspond to much longer atmospheric lifetimes (6.2 and 6.4 days for α- and β-TBECH, respectively, using COSMOtherm; 2.5 days for both isomers using AOPWIN) than those predicted by Wang et al. (2021). These longer lifetimes and the similar rates of reaction of the two main isomers are more consistent with the observed atmospheric dispersion of TBECH.**

Line 437, Section 4.3. It seems that the "elephant in the room" has been ignored in this discussion – the possibility of air transport from the U.S.! Especially since the sampling sites are not far from the border. What is known about TBECH use south of the border?

TBECH is listed in EPA's Toxic Substances Control Act (TSCA) Chemical Substance Inventory (last updated: February 2023). Its commercial activity status is currently marked as "inactive," indicating that there is no manufacturer or processor in the US that produces or imports TBECH for non-exempt commercial purposes (https://www.epa.gov/tsca-inventory/tsca-inventory-notification-active-inactive-rule). Moreover, TBECH currently has no regulatory flags in the inventory, most likely resulting in low priority for environmental assessment. In other words, the situation with TBECH in the US is similar to that of in Canada, where according to official documentation, TBECH should not exist in the US. Therefore, atmospheric transport from point sources in the US is not likely.

If atmospheric transport from the US played a contributory role in the amount of TBECH detected in the Canadian atmosphere, then we would also expect a more uniform spatial variability in the air concentration of TBECH in Canada. The distinct relationship between air concentration and population observed in our study suggests that the main source of TBECH is domestic.

Section 4.3 will be changed to include the US in the discussion: "The inventory update phase 3 survey in 2017 under Canada's Chemical Management Plan (CMP) resulted in no reports of manufacturing or importing TBECH into the country by domestic companies, eliminating the possibility of point sources located in Canada for this flame retardant, such as factories that produce or import commercial TBECH. **Similarly, the United States Environmental Protection Agency's (US EPA) Toxic Substances Control Act (TSCA) Chemical Substance Inventory contains no records of the production or importation of TBECH for commercial purposes in the US, indicating that there are also no point sources of TBECH located in the US to contribute to the TBECH levels observed in Canada *via* atmospheric transport."**

---

## Author Comment (AC2)

**Response to Anonymous Reviewer #2's referee comment on: Oh, Jenny, The atmospheric fate of 1,2-Dibromo-4- (1,2-dibromoethyl)cyclohexane (TBECH…) (egusphere-2023-1151)**

Keywords: I did not actually see the keywords, but consider including the other name to TBECH there: DBE-DBCH

As the journal ACP appears to not require keywords, the manuscript will be modified to include DBE-DBCH in the abstract. Changes are made in bold: "Here, we report the concentration of the α- and β-isomers of 1,2-dibromo-4-(1,2-dibromoethyl) cyclohexane (TBECH**; also known as DBE-DBCH**) in over 300 air, water, and precipitation samples collected between 2019 and 2022 using active air and deposition sampling as well as networks of passive air and water samplers."

Line 50 - it could be mentioned that toxicity & effects can still occur with non-persistent contaminants - if they are continuously released.

Thank you very much for the feedback. While we share the opinion by the reviewer, we feel that it is not necessary to add a sentence on this matter.

Line 71-74 - There are a few other toxic effects - you could simply point readers to a recent review which covers many of them (Marteinson et al 2021: A review of 1,2-dibromo-4-(1,2-dibromoethyl)cyclohexane in the environment and assessment of its persistence, bioaccumulation and toxicity, Environmental Research, 195.)

Thank you for providing us with a reference to this comprehensive review on TBECH. We will modify the introduction to include the reference when describing the health effects of TBECH: "This disrupting potential of TBECH is also seen in the *in vivo* studies reporting changes in circulating hormones (Curran et al., 2017; Gemmill et al., 2011), organ structure (Park et al., 2011), and reproduction (Marteinson et al., 2012a; Marteinson et al., 2012b) after exposure to low concentrations of TBECH. **More details on the reported toxicological effects of TBECH from both *in vivo* and *in vitro* studies are discussed in Marteinson et al. (2021).**"

102 - 2019 - 22 - should any potential impacts of change human behaviour during the pandemic be mentioned? I am not sure what these might be given TBECH is likely emitted from finished imported products. However, it does come to the reader's mind that this might not be a representative period of time. There is no need to speculate, but it could simply be acknowledged a an unknown; perhaps there is some trend data on other similar air pollutants to indicate any changes (or not) during the pandemic that could be mentioned?

While the COVID-19 pandemic greatly disrupted anthropogenic activity worldwide, we do not believe that this would cause TBECH emissions to change greatly. Emission of TBECH to the atmosphere is not dependent on human behaviour. Therefore, products containing TBECH (e.g., electronics, plastics) already existing in Canada would simply continue to leak TBECH during the pandemic. Other atmospheric contaminants that showed elevated/decreased levels during the lockdown, such as $O_3$, $NO_2$, and NO are tied to pollution sources that saw drastic decrease in activity during the lockdowns, such as using vehicles and running factories (https://doi.org/10.1016/j.scitotenv.2020.140516).

Line 181-183 - I think it would be beneficial to summarize in a few lines what these tables show - otherwise the results for all of these other BFRs measured are only found in the supplemental information and that is unfortunate.

Thank you for the feedback. We have modified the manuscript to include a short description about the other BFRs we analyzed: "To a lesser extent, several other EBFRs and PBDE congeners were also detected in the samples of this study, with their detection frequencies or concentrations summarized in Tables S4 to S8. **Several legacy BFRs (BDE-190 and 85) and EBFRs (HBBz and PBBz) were occasionally detected in the PAS network samples, with detection frequencies ranging between 7 to 26%. These compounds, however, were not detected in the active air samples. Moreover, other BFRs that were detected frequently in water samples (BDE-17, 28, 99, and 100) were rarely detected in the other sample types.** Because of their much higher detection frequencies **in all samples**, the remainder of the manuscript is focused on TBECH and BDE-47."

Paragraph beginning line 185 - Note that concentrations of TBECH have already been reviewed (Marteinson et al 2020) which includes most of these same references - this should be acknowledged.

Thank you for pointing this out to us. Section 3.1.1 of the manuscript will be changed to include the given reference: "To place these concentrations into context, Table S13 summarises all atmospheric concentrations of TBECH previously reported in the literature. **A review of the occurrences of TBECH in other environmental media (e.g., soil, water, sediment) can be found in Marteinson et al. (2020)."**

Line 204-5 "in this study, BDE-47 was also detected in the air in all sampling regions, with comparable levels to TBECH." This is a very significant and important finding, adding important evidence to that mounting which suggests TBECH is an contaminant of concern to consider further in risk assessment.

We appreciate that the reviewer agrees with our conclusion that TBECH warrants further consideration for risk assessment.

The figures are excellent.

We are grateful for the positive feedback.

Line 318 - can you verify this error before publication?

Before the submission of the manuscript, we had notified the organizers of the Federal Whale Initiative about this possible error. We have not received a confirmation, but will delete the phrase in brackets if this should be resolved prior to publication: "The median water grab sampling concentrations of TBECH (sum of all isomers) and BDE-47 reported by the FWI were 22.9 pg/L **(presumably incorrectly labelled as ng/L in the database)** and 32 pg/L, respectively, which are one to two orders of magnitude higher than the PWS measurements."

Paragraph beginning on line 354 - This is also a very interesting/novel finding and useful for understanding of TBECH enantiomers in the environment.

We also agree that the results of the enantiomeric fraction analysis of TBECH in the environment is novel. We appreciate the favourable comment to our results.

Section 4.1.2 - It was striking to me that air concentrations of TBECH changed with the season, but precipitation concentrations did not - some discussion of this here and how it relates to the scavenging rations you have calculated would be beneficial/interesting.

Thank you very much for the suggestion. We will modify section 4.1.2 to include a paragraph on these observations and its relation to scavenging ratios:"**The month-to-month variability of the precipitation concentrations were also too large to reveal a clear seasonal pattern. This suggest that variability of TBECH levels in precipitation is controlled by factors that differ from those that control seasonal variability in air concentrations (e.g., temperature and air mass origin). Potential candidates for those factors are related to the nature of the precipitation events (e.g., frontal vs. convective storms, snow vs. rain, and precipitation rate). However, another phenomenon could also occur: Higher temperatures in summer favour higher air concentrations but lower the precipitation scavenging efficiencies of vapours, due to the temperature dependence of $K_{WA}$. This might explain why concentrations in precipitation do not peak in summer even if concentrations in air do.**"

4.2 - this is the only heading phrased as a question which seems out of place - you may want to change it.

We titled section 4.2 as one of the main questions we were aiming to answer with our results. There are rarely any studies on the environmental fate of TBECH, and even less on its isomers. With so little prior information available, we believe that leaving the title as a question highlights the novelty of its answer.